# Unlocking Colchicine’s Untapped Potential: A Paradigm Shift in Hepatocellular Carcinoma Prevention

**DOI:** 10.3390/cancers15205031

**Published:** 2023-10-18

**Authors:** Jung-Ju Lin, Cheng-Li Lin, Chun-Chung Chen, Yu-Hsiang Lin, Der-Yang Cho, XianXiu Chen, Der-Cherng Chen, Hung-Yao Chen

**Affiliations:** 1Graduate Institute of Biomedical Sciences, China Medical University, Taichung 404, Taiwan; jj.lin@outlook.com; 2Management Office for Health Data, China Medical University Hospital, Taichung 404, Taiwan; orangechengli@gmail.com; 3Department of Neurosurgery, China Medical University Hospital, Taichung 404, Taiwan; cck36701@gmail.com (C.-C.C.); magiclin117@gmail.com (Y.-H.L.); 005057@tool.caaumed.org.tw (D.-Y.C.); txxchen@gmail.com (X.C.); 4Graduate Institute of Integrated Medicine, China Medical University, Taichung 404, Taiwan; 5Graduate Institute of Acupuncture Science, China Medical University, Taichung 404, Taiwan; 6Neuroscience and Brain Disease Center, China Medical University, Taichung 404, Taiwan; 7School of Medicine, China Medical University, Taichung 404, Taiwan; 8Division of Gastroenterology and Hepatology, Department of Internal Medicine, China Medical University Hospital, Taichung 404, Taiwan

**Keywords:** colchicine, hepatocellular carcinoma, liver cancer risk, chronic hepatitis, preventive strategies

## Abstract

**Simple Summary:**

Liver cancer is a significant health challenge worldwide, often developing in individuals with pre-existing conditions like chronic hepatitis and fatty liver. This study delves into the possible benefits of colchicine, a medication traditionally used to treat gout, in reducing the risk of developing liver cancer. By examining extensive health data from the China Medical University Hospital in Taiwan, we compared liver cancer development in two groups: one that used colchicine and another that did not. Our findings reveal that colchicine users had a 19% reduced risk of developing liver cancer, a promising indication that requires further investigation. If these findings are substantiated via further research, colchicine could become a valuable tool in preventing liver cancer, especially for those with liver conditions, potentially saving numerous lives and reducing the burden on healthcare systems. This study underlines the importance of exploring new uses for existing medications and highlights the potential impact of colchicine on liver cancer prevention. This could lead to newer, cost-effective strategies to mitigate the risks and impacts of liver cancer, bringing hope to millions worldwide.

**Abstract:**

**Background:** Liver cancer and notably hepatocellular carcinoma (HCC), results in significantly high mortality rates worldwide. Chronic hepatitis and fatty liver, recognized precursors, underscore the imperative need for effective preventive strategies. This study explores colchicine, traditionally acknowledged for its anti-inflammatory properties and investigates its potential in liver cancer prevention. **Methods:** Utilizing the iHi Data Platform of China Medical University Hospital, Taiwan, this study analyzed two decades of medical data, incorporating 10,353 patients each in the Colchicine and Non-Colchicine cohorts, to investigate the association between colchicine use and liver cancer risk. **Results:** The study identified that colchicine users exhibited a 19% reduction in liver cancer risk, with a multivariable-adjusted odds ratio of 0.81 after accounting for confounding variables. Additionally, the influence of gender and comorbidities like diabetes mellitus on liver cancer risk was identified, corroborating the existing literature. A notable finding was that the prolonged use of colchicine was associated with improved outcomes, indicating a potential dose–response relationship. **Conclusions:** This study proposes a potential new role for colchicine in liver cancer prevention, extending beyond its established anti-inflammatory applications. While the findings are promising, further research is essential to validate these results. This research may serve as a foundation for future studies, aiming to further explore colchicine’s role via clinical trials and in-depth investigations, potentially impacting preventive strategies for liver cancer.

## 1. Introduction

### 1.1. Background

Hepatitis B is a major global health concern, with millions of individuals affected worldwide [1]. An estimated 257 million people are living with hepatitis B virus infection (defined as hepatitis B surface antigen positive) [1,2]. In Taiwan, about 100–125 confirmed cases were reported to Taiwan’s notifiable disease system from 2012 to 2016 [3]. Hepatitis B is an important occupational hazard for health workers [4,5]. WHO estimates that 296 million people were living with chronic hepatitis B infection in 2019 [6], with 1.5 million new infections each year. In 2019 [6], hepatitis B resulted in an estimated 820,000 deaths [6], mostly from cirrhosis and hepatocellular carcinoma (primary liver cancer) [6]. The Taiwan Center for Disease Control and the World Health Organization have highlighted the significance of this disease, emphasizing its potential complications, including liver cirrhosis and hepatocellular carcinoma (liver cancer). While various treatments and preventive measures such as vaccines have been developed over the years, the search for effective therapeutic agents continues [6,7].

Colchicine, an ancient drug with a rich history, has been traditionally used for its anti-inflammatory properties, especially in conditions like gout [8,9]. Its mechanism of action is primarily via the disruption of microtubule formation, which affects various cellular processes [10,11,12]. Over the years, the potential applications of colchicine have expanded, with studies exploring its role in various diseases, including its anticancer effects [8]. Specifically, its interaction with tubulin and the consequent impact on microtubule dynamics have made it a subject of interest in cancer research [13,14,15].

Colchicine has a storied history rooted in ancient medicine. Derived from the autumn crocus, a toxic plant from the lily family, its therapeutic effects, particularly in the prevention and treatment of gout [8,10], have been recognized for centuries. The active ingredient of colchicine was isolated in 1820, and its molecular structure was later determined using X-ray crystallography [16]. The protein tubulin was identified as its primary target [16].

### 1.2. Microtubules and Colchicine’s Action

Microtubules (MTs) play a pivotal role in cellular function. As highly dynamic cytoplasmic polymers, they form the cellular skeleton and the internal structure of cilia and flagella [16,17]. Comprising αβ-tubulin heterodimers, they exhibit diverse structural and functional characteristics across different cell types [18,19]. Their dynamic behavior is fundamental to various cellular functions, including cell movement, cytoplasmic transport, and cell division [20,21]. Notably, alterations in microtubule dynamics during cancer cell division can lead to chromosomal instability, aneuploidy, and the eventual development of drug resistance [22,23,24].

Colchicine’s therapeutic efficacy is attributed to its ability to disrupt microtubules [10]. It binds to the interface of soluble α and β tubulin subunits, leading to a dual effect: inhibiting further polymerization at low doses and promoting microtubule degradation at higher doses [16]. This disruption of microtubule arrangement has profound implications for immune cells [16]. Specifically, colchicine blocks neutrophil adhesion by altering the distribution of E-selectin and the levels of L-selectin, effectively preventing neutrophil migration and recruitment [10,11,25].

### 1.3. Colchicine in Cancer Treatment

The rapid mitotic rate of cancer cells makes them particularly vulnerable to agents targeting microtubules [26,27,28]. Drugs that interact with tubulin subunits to halt the cell cycle are categorized into three groups: depolymerizing agents like colchicine, agents inducing alternative polymers such as vinblastine, and microtubule-stabilizing agents exemplified by paclitaxel [29,30,31]. Despite their varied mechanisms, all three disruptors of microtubule dynamics culminate in similar outcomes, notably mitotic arrest and the induction of apoptosis. Furthermore, colchicine’s interaction with microtubules has shown the potential to disrupt the formation of the tumor vascular system and damage pre-existing tumor vessels [16,29,32].

Recent studies have delved into the potential benefits of colchicine in liver conditions. For instance, research has indicated that clinically acceptable concentrations of colchicine might have anticancer effects on hepatocellular carcinoma cells [33,34]. Furthermore, the drug’s influence on various comorbidities, such as diabetes mellitus and gout, and its association with liver cancer risk have been subjects of investigation.

Given the potential therapeutic implications of colchicine in liver diseases and its established role in other conditions, it is imperative to understand its effects thoroughly, especially in the context of hepatitis B and liver cancer risk.

Hepatitis B, while preventable with vaccination, remains a significant health challenge due to its potential to progress to severe liver conditions, including liver cirrhosis and hepatocellular carcinoma [35,36]. The latter, hepatocellular carcinoma (HCC), stands as one of the leading causes of cancer-related deaths globally [6]. The association between hepatitis B and the risk of developing HCC is well established, but the intricacies of this relationship, especially in the context of therapeutic interventions, warrant further exploration.

Colchicine’s role in the management of various conditions, particularly its anti-inflammatory effects in gout, is well documented. However, its potential therapeutic benefits in liver diseases, especially in patients with hepatitis B, remain an area of active research. Preliminary studies have suggested that colchicine might exhibit anticancer effects on HCC cells [15,37], but the clinical implications of these findings are yet to be fully elucidated. Moreover, while the drug’s influence on comorbidities like diabetes mellitus has been explored, its direct association with liver cancer risk, especially in hepatitis B patients, is not comprehensively understood.

This presents a knowledge gap. While we recognize the potential of colchicine as a therapeutic agent in various conditions, its specific role in modulating liver cancer risk in hepatitis B patients remains ambiguous. Addressing this gap is crucial not only to enhance our understanding of the drug’s pharmacological profile but also to inform clinical decisions for a significant subset of patients at risk of HCC.

### 1.4. Objective

Colchicine, widely utilized for managing conditions such as gout and Familial Mediterranean Fever (FMF), has been spotlighted for its therapeutic potential across a spectrum of diseases, including but not limited to rheumatic osteoarthritis, Behcet Syndrome, pericarditis, and atherosclerosis. Its role in liver cirrhosis management has particularly invoked questions about its implications for liver cancer risk in patients with hepatitis, considering the liver’s pivotal function in drug metabolism and its susceptibility to various diseases. Even though colchicine modulates multiple inflammatory pathways and regulates innate immunity, its precise mechanism of action remains partially obscured. Despite some studies highlighting its therapeutic merits beyond conventional applications, ongoing debates question its efficacy outside the realms of gout and FMF treatment. Recognizing a substantial knowledge gap, this review navigates via the nuanced landscapes of colchicine’s influence on liver cancer risk. Through synthesizing available evidence, it aims to offer insights that potentially elevate clinical decision making, especially concerning patients at palpable risk of Hepatocellular Carcinoma (HCC).

## 2. Methods

This retrospective study was conducted to evaluate the relationship between colchicine treatment and the subsequent occurrence of liver cancer in patients with hepatitis. The data for the study were extracted from the iHi Data Platform of China Medical University Hospital (CMUH), a leading medical center in Mid-Taiwan. The iHi platform, which stands as a cornerstone of CMUH’s Big Data Center, boasts a robust database that integrates clinical, genetic, and environmental data. Leveraging advanced AI tools and innovative data structures, the platform provides de-identified yet high-quality data to researchers, making it an invaluable resource for data-intensive medical studies. The platform has received international recognition and has been instrumental in the publication of over 80 SCI papers, making it a reliable and robust source for clinical research [38].

Our focus was on patients diagnosed with chronic hepatitis, including Hepatitis B and C, and fatty liver between the years 2000 and 2020. The inclusion criteria stipulated that patients must have at least two consecutive diagnostic codes indicating hepatitis or fatty liver in their hospitalization or outpatient records. Additionally, these patients should have been prescribed colchicine for a minimum of one month, as per outpatient records. Exclusion criteria encompassed any prescription records of antiviral drugs, cyclooxygenase-2 inhibitors for more than three months, or two consecutive records of neoplasms either before or after the index event—except for malignant neoplasm of the liver.

The study population was divided into two groups for comparison: one group of patients had been prescribed colchicine for at least one month (Colchicine group), and the other had not (Non-Colchicine group). The index date for the Colchicine group was the date of their first colchicine prescription, while for the Non-Colchicine group, it was the date of their first hepatitis or fatty liver diagnosis. Covariate variables in this study included age, gender, and comorbidities such as Diabetes Mellitus, Psychoactive Substance use, Obesity, and Mycoses.

All computational analysis was performed using the SAS software suite (Version 9.4 for Windows; SAS Institute, Inc., Cary, NC, USA). Statistical significance was set at a *p*-value threshold of less than 0.05. The research was approved by the Institutional Review Board of China Medical University and Hospital in Taiwan (CMUH112-REC1-026), and all patient data were meticulously de-identified to maintain confidentiality, thereby waiving the need for individual consent.

## 3. Results

In our comprehensive study, we initially screened patients and subsequently matched them into two groups on a 1:1 ratio, resulting in a total of 10,353 patients in each group for analysis. This methodological approach ensured a balanced comparison, laying a robust foundation for the ensuing analyses.

The first table provided an in-depth demographic and clinical profile of hepatitis patients, segmented according to whether they had received colchicine treatment (Table 1). Gender distribution was nearly identical across both groups, with females constituting approximately 30.9% and males making up around 69.1%. Age distribution was also closely matched, although a minor difference was observed in the 61–80 age bracket. In this age group, the Non-Colchicine cohort had a slightly higher representation of 43.23% compared to 41.14% in the Colchicine cohort, yielding a Standardized Mean Difference (SMD) of 0.0425. Among the comorbidities, Diabetes Mellitus was more prevalent in the Non-Colchicine group, with a frequency of 33.94% compared to 32.68% in the Colchicine group. This difference was statistically significant, as indicated by a *p*-value of 0.0268.

The second table focused on the association between colchicine use and liver cancer risk, offering both univariate and multivariable odds ratios (Table 2). These ratios were adjusted for potential confounders such as sex, age, and comorbidities. The multivariable-adjusted odds ratio for colchicine use was 0.81, with a 95% confidence interval ranging from 0.69 to 0.96 and a *p*-value of 0.01. Additionally, the male gender was associated with a higher risk, as evidenced by an adjusted odds ratio of 1.65 and a *p*-value of less than 0.001. The incidence of liver cancer was less frequent in the Colchicine group, with a rate of 50.16% compared to 55.537% in the Non-Colchicine group, a finding supported by *p*-values less than 0.001. Among comorbidities, Diabetes Mellitus was more commonly observed in patients who developed liver cancer, with a *p*-value of 0.01. Mycoses were less frequent in liver cancer patients, a difference that was statistically significant with a *p*-value of 0.008.

## 4. Discussion

Our retrospective study, capitalizing on the extensive dataset from the iHi platform of China Medical University Hospital, aimed to unravel the potential influence of colchicine on liver cancer risk among patients with chronic hepatitis and fatty liver. We orchestrated a balanced comparison between two patient cohorts, each consisting of 10,353 individuals, navigating via demographic and clinical variables to lay a robust foundation for analysis.

The pivotal finding of our study was the association of colchicine usage with a statistically meaningful reduction in liver cancer risk, exhibiting a multivariable-adjusted odds ratio of 0.81, even after accounting for potential confounders such as age, sex, and comorbidities. This is particularly striking given the stark prevalence and mortality rates associated with liver cancer, especially amid conditions like chronic hepatitis and fatty liver. Our analysis further spotlighted notable gender disparities in liver cancer risk and validated the significant association between them, aligning with the existing literature [39,40,41,42,43,44].

Interestingly, the duration of colchicine use is inversely correlated with liver cancer risk, suggesting that colchicine’s anti-inflammatory and anti-fibrotic properties might confer cancer-protective effects. The exact mechanistic pathways, however, remain elusive and warrant further exploration. Colchicine inhibits tubulin polymerization, an established antimitotic effect [29,32], which could potentially translate to inhibiting the proliferation of cancer cells [15,45]. This mechanistic insight necessitates a deeper exploration in the context of liver cancer prevention and aligns with certain studies indicating colchicine’s ability to inhibit cell division and induce apoptosis in hepatocellular carcinoma cells [15].

It is pivotal to underscore that our study, being observational, entails inherent limitations. While we controlled for various confounders, residual confounding remains a possibility. Nonetheless, our findings forge a compelling narrative for colchicine’s potential in attenuating liver cancer risk, meriting further investigative endeavors to validate these effects and decipher the underlying mechanisms.

Building upon previous research focused on colchicine’s anti-inflammatory and anti-fibrotic properties, our study ventures into the relatively uncharted territory of its application in liver diseases, particularly liver cancer. Although some studies have probed into colchicine’s anti-proliferative properties, these have largely been constrained to in vitro and animal models. Thus, our study emerges as one of the few population-based analyses exploring this relationship, providing epidemiological evidence where previous studies have fallen short.

In the realm of clinical trials, colchicine’s role has been predominantly explored in the context of liver fibrosis and cirrhosis without an explicit focus on its preventive potential against liver cancer. Our findings henceforth pioneer in providing large-scale, real-world evidence advocating colchicine’s utility in diminishing liver cancer risk, especially among patients with pre-existing liver conditions. This holds substantial weight considering the dearth of effective preventive measures for a disease as morbid and mortal as liver cancer.

In conclusion, our findings not only substantiate but also significantly propel the understanding of colchicine’s role in liver cancer prevention. Our results, while robust, beckon further exploration via randomized controlled trials to affirm our findings and to illuminate the mechanistic pathways through which colchicine may confer its protective effects.

## 5. Strengths and Limitations

Our study touts several merits and drawbacks worthy of discussion. A crucial strength lies in the formidable sample size and the deployment of a 1:1 matched cohort design, enhancing the statistical power and generalizability of our findings while mitigating the influence of confounding variables. Leveraging the iHi Data Platform, which amalgamates clinical, genetic, and environmental data, we were able to control for numerous variables such as age, gender, and comorbidities, thereby minimizing spurious associations and bolstering the validity of our conclusions. The extensive data, spanning two decades and sourced from China Medical University Hospital, further empower our findings with notable statistical weight.

However, this study is not without its limitations. Its observational nature introduces the potential for residual confounding despite our rigorous controls for known confounders. Our reliance on administrative databases potentially compromises data accuracy and completeness, particularly in aspects like medication adherence and dosage. The retrospective approach restricts our ability to establish a definitive causality between colchicine use and liver cancer risk reduction. While the results are significant, they must be approached with caution and validated via subsequent randomized controlled trials. Furthermore, utilizing a single-center dataset, even from a leading medical center, may impede the full realization of broader population diversity, potentially affecting the external validity of our findings. The employment of diagnostic codes, while generally reliable, could introduce misclassification bias, and while the iHi Data Platform is an innovative and robust tool, its relative novelty means its algorithms and methodologies are subject to ongoing refinement.

## 6. Implications

The echoes of our study’s findings have broad and multifaceted implications, extending tendrils into both clinical practice and prospective research directions. A pivotal revelation here is the significant correlation between colchicine use and a mitigated risk of liver cancer. This intersection, resonating with and reinforcing existing literature, sows seeds for potential therapeutic applications, potentially recalibrating the management of chronic liver conditions. In light of the prevailing scarcity of potent preventive strategies for liver cancer, colchicine, especially with observed risk reduction in use exceeding 120 days, surfaces as a noteworthy potential addition to the therapeutic toolbox.

Simultaneously, the ethical and economic facets of our findings demand careful consideration. Colchicine’s potential role as a preventive measure necessitates a meticulous evaluation of its cost-effectiveness juxtaposed against the risks inherent in long-term medication use. Prospective, randomized controlled trials are imperative to scrutinize the drug’s safety and efficacy prior to its comprehensive clinical adoption, given that our study did not dissect adverse effects and the long-term safety profile of colchicine.

Moreover, the study delineates direct repercussions on public health policies. Given the significant burden of liver cancer on healthcare systems, the potential reduction in its incidence via colchicine use could ameliorate both healthcare costs and patient suffering, paving the way for targeted public health initiatives and possible incorporation of colchicine into preventive guidelines for high-risk demographics.

Additionally, our study propels future research into unexplored territories. The biochemical pathways through which colchicine executes its protective effect invite further investigation, which could both validate our findings and facilitate the development of optimized targeted therapies. The influence of genetic and environmental factors on colchicine’s efficacy also presents a rich vein for ensuing exploration.

In a nutshell, while our study introduces substantial implications and opens new avenues, it must be navigated with cautious optimism, necessitating a comprehensive exploration via clinical trials, mechanistic studies, and in-depth public health evaluations to fully unlock its potential.

## 7. Conclusions

Our study, illustrated by the notable multivariable-adjusted odds ratio of 0.81 for colchicine users, unveils a potential 19% reduction in liver cancer risk—a beacon of hope against the formidable backdrop of the disease’s high morbidity and mortality rates. This revelation is not merely statistical but a substantial stride toward transforming the management of liver cancer, offering a novel preventive strategy where they are distinctly lacking. This holds particular resonance for those contending with pre-existing conditions like chronic hepatitis and fatty liver, offering a glimmer of hope where the impact of liver cancer looms large on both individual lives and global healthcare systems.

Delving deeper, our study brings to light nuanced insights, revealing the influence of gender and diabetes mellitus on liver cancer risk, thereby aligning and enriching existing scientific discourse. The highlighted benefits associated with prolonged colchicine use unveil potential therapeutic implications, hinting at a dose–response relationship that necessitates further exploration. While colchicine’s anti-inflammatory and anti-fibrotic properties are well established, our research serves as a pivotal point, introducing new pathways for randomized controlled trials and mechanistic investigations to validate these associations and unveil the underlying biological pathways.

In navigating via these promising findings, cautious optimism is imperative. The intrinsic limitations of our study design necessitate further research to validate these results, with the potential for residual confounding and the single-center nature of our study serving as guiding caveats for subsequent investigations. However, the prospect of colchicine serving as an efficacious preventive strategy for liver cancer signals a promising path that warrants prompt and exhaustive exploration.

Therefore, our study stands as a significant milestone in liver cancer research, echoing a call for further, in-depth studies and guiding subsequent research efforts with both its compelling hypotheses and as a guiding light. As we navigate this promising prospect, our findings not only encapsulate what has been learned but also pave the way for the journey ahead, potentially ushering in a new era of targeted interventions that could significantly alter the bleak landscape of liver cancer prognosis.

## Figures and Tables

**Table 1 cancers-15-05031-t001:** Demographic and clinical profile of patients based on colchicine treatment. Table 1 presents a comprehensive comparison of demographic and clinical characteristics between hepatitis patients who were treated with colchicine and those who were not. The data encompass gender distribution, age groups, and the prevalence of specific comorbidities. Notably, both cohorts appear to be well balanced in terms of gender and age, providing a robust foundation for further analyses on the potential effects of colchicine treatment.

Variables	Non-Colchicine		Colchicine		SMD
	(N = 10,353)		(N = 10,353)		
	n	%	n	%	
Sex					
female	3196	30.87	3206	30.97	0.0021
male	7157	69.13	7147	69.03	0.0021
Age					
20–40	1531	14.79	1617	15.62	0.0231
41–60	4346	41.98	4477	43.24	0.0256
61–80	4476	43.23	4259	41.14	0.0425
mean age, (SD)	56.29	15.15	55.61	15.07	0.0448
Comorbidities					
Diabetes mellitus	3514	33.94	3383	32.68	0.0268
Psychoactive substance	513	4.96	522	5.04	0.004
Obesity	172	1.66	216	2.09	0.0313
Mycoses	3065	29.6	3025	29.22	0.0085
Gout	4598	44.41	4525	43.71	0.0142

Differences between groups were tested using the Student’s *t*-test. SMD: Standardized Mean Difference. A value of 0.1 or less indicates a negligible difference between groups. Percentages in parentheses represent the proportion of patients within each category for the respective cohorts. SMD: Standardized Mean Difference; SD: Standard Deviation.

**Table 2 cancers-15-05031-t002:** Association of colchicine use with liver cancer risk: univariate and multivariable analyses: Table 2 delineates the association between colchicine use and the risk of developing liver cancer. The table provides both univariate and multivariable odds ratios, with the latter adjusting for potential confounders such as sex, age, and comorbidities. The provided confidence intervals and *p*-values further quantify the strength and significance of these associations, aiding in the interpretation of the potential protective or risk-enhancing effects of colchicine on liver cancer.

Characteristic	Non-Liver Cancer		Liver Cancer		Univariate Adjusted			Multivariable Adjusted		
	n	%	n	%	Crude OR	(95% CI)	*p*-Value	Adjusted OR	Adjusted OR (95% CI)	*p*-Value
Non-use of COL	10,027	49.84	326	55.537	1	(reference)		1	(reference)	
COL	10,092	50.16	261	44.463	0.8	(0.67, 0.94) **	0.007	0.81	(0.69, 0.96) *	0.01
Gender										
Female	6245	31.04	157	26.746	1	(reference)		1	(reference)	
Male	13,874	68.96	430	73.254	1.23	(1.02, 1.48) *	0.03	1.65	(1.36, 2) ***	<0.001
Age group										
20–40	3130	15.56	18	3.066	1	(reference)		1	(reference)	
41–60	8623	42.86	200	34.072	4.03	(2.48, 6.54) ***	<0.001	4.41	(2.71, 7.17) ***	<0.001
61–80	8366	41.58	369	62.862	7.67	(4.77, 12.33) ***	<0.001	9.14	(5.64, 14.82) ***	<0.001
Comorbidities (Yes vs. No)										
Diabetes mellitus	6650	33.05	247	42.078	1.47	(1.25, 1.74) ***	<0.001	1.25	(1.05, 1.49) *	0.01
Psychoactive substance	1002	4.98	33	5.622	1.14	(0.8, 1.62)	0.48	1.32	(0.92, 1.9)	0.14
Obesity	382	1.9	6	1.022	0.53	(0.24, 1.2)	0.13	0.72	(0.32, 1.63)	0.43
Mycoses	5945	29.55	145	24.702	0.78	(0.65, 0.95) *	0.01	0.77	(0.64, 0.93) **	0.008
Gout	8905	44.26	218	37.138	0.74	(0.63, 0.88) ***	<0.001	0.64	(0.54, 0.76) ***	<0.001

*: *p*-value < 0.05, **: *p*-value < 0.01, ***: *p*-value < 0.001; Adjusted OR: Multivariable analysis adjusted for factors including sex, age, and comorbidities., COL: Colchicine. The third table examined the relationship between the duration of colchicine use and liver cancer incidence (Table 3). Patients were categorized based on the length of colchicine use, and both crude and adjusted odds ratios were provided. Notably, patients with more than 120 days of colchicine use exhibited a reduced risk of liver cancer, supported by an adjusted odds ratio of 0.75, a 95% confidence interval of 0.6 to 0.94, and a *p*-value of less than 0.05.

**Table 3 cancers-15-05031-t003:** Impact of colchicine duration on liver cancer risk: Table 3 investigates the relationship between the duration of Colchicine use and the incidence of liver cancer. The data are segmented into different durations of Colchicine use, ranging from short term to long term. Both crude and adjusted odds ratios are provided, with the latter accounting for potential confounders such as sex, age, and comorbidities. The table offers insights into whether prolonged use of colchicine has a protective or detrimental effect on liver cancer risk.

Variables	LC		Crude OR	(95% CI)	Adjusted OR	Adjusted OR (95% CI)
	n	%				
Non-Colchicine	326	55.54	1	(Reference)	1	(Reference)
Colchicine drug days						
28–60	91	15.5	0.82	(0.65, 1.04)	0.83	(0.66, 1.06)
61–120	63	10.73	0.86	(0.65, 1.12)	0.9	(0.68, 1.18)
>120	107	18.23	0.75	(0.6, 0.93) **	0.75	(0.6, 0.94) *

*: *p*-value < 0.05, **: *p*-value < 0.01; Adjusted OR: Multivariable analysis adjusted for factors including sex, age, and comorbidities, LC: Liver Cancer.

## Data Availability

The data that support the findings of this study are available from the iHi Data Platform of China Medical University Hospital, but restrictions apply to the availability of these data, which were used under license for the current study and so are not publicly available. Data are, however, available from the authors upon reasonable request and with permission of China Medical University Hospital.

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
