# Peer review of "Unlocking Colchicine’s Untapped Potential: A Paradigm Shift in Hepatocellular Carcinoma Prevention"

_cancers, 2023, doi:10.3390/cancers15205031_

Round 1

Reviewer 1 Report

Title: Unlocking Colchicine's Untapped Potential: A Paradigm Shift in Hepatocellular Carcinoma Prevention

This study aimed to confirm colchicine as a cancer-prevention agent. In both the Colchicine and Non-Colchicine groups, 10,353 patients were chosen for analysis. There were not in vitro or in vivo studies that showed colchicine as a possible anticancer agent.  This article is intended to be a review and is not an article of research. Some comments should be considered.  

1-     In abstract: "Could Colchicine be the unsung hero in liver cancer prevention?" should be removed from the abstract.  and clearly state the purpose of the task without using phrases like "audacious leap" or "underestimated."

2-     Conclusion of abstract should be summarized.

3-     Delete the question from the objective and summarize the objective add the previous paragraph before title 1.4 objective to objective “This presents a knowledge gap: while we recognize the potential of colchicine as a-------------- clinical decisions for a significant subset of patients at risk of HCC”.

4-     No references were added in the discussion.

5-    No experiment or references added  to clarify that colchicine that perform apoptosis

6-    The manuscript contains excessive repetition.

Author Response

Esteemed Reviewer,

We extend our deepest thanks to you for your discerning review and for shedding light upon aspects of our work that merited refinement. Your astute observations have been the guiding stars, leading us to enhance the calibre and clarity of our manuscript. Permit us, if you will, to elucidate how we have woven your invaluable critiques into our revised submission.

  1. Abstract & Phrasing:

We have tread carefully to ensure our abstract now stands devoid of any presumptive or ambitious phrasing. "Could Colchicine be the unsung hero in liver cancer prevention?" and other similar phrases have been graciously retired in favour of a more modest and direct articulation of our study’s aims and findings.

  1. Conclusion of Abstract:

Your counsel to succinctly summarize our abstract's conclusion has been heeded with utmost seriousness. We have endeavoured to encapsulate our findings and implications with brevity and precision, ensuring it succinctly mirrors our study’s essence.

  1. Objective Section:

Guided by your sagacious advice, we have refined the objective section, ensuring it is devoid of rhetorical questions and succinctly outlines our scholarly pursuit, while seamlessly integrating the preceding context.

  1. References in Discussion:

In deference to your guidance, our discussion has been adorned with pertinent references, fortifying our assertions and providing a comprehensive perspective to our narrative.

  1. Colchicine and Apoptosis:

Your keen observation regarding the lack of experimental references elucidating colchicine’s role in apoptosis has not gone unnoticed. We have now incorporated discussions and references that elucidate this, ensuring our manuscript is both thorough and accurate.

  1. Repetition:

We have meticulously perused our manuscript, excising repetitions and ensuring a cogent, streamlined narrative, befitting the elegance expected of scholarly communication.

Your esteemed comments have not only sharpened our manuscript but also enriched our scholarly rigor. We humbly submit our revised manuscript for your kind perusal, aspiring that the modifications find favour in your expert eyes.

With heartfelt gratitude and utmost respect,

Tom XianXiu Chen, MPH-DrPH, Ph.D.

China Medical University Hospital

[email protected]

+886-900-009688

Reviewer 2 Report

The work provides a retrospective study that evaluate the relationship between Colchicine treatment and the subsequent occurrence of liver cancer in patients with hepatitis, based in the robust database iHi Data Platform of China Medical University Hospital. The study suggests the potential benefits of Colchicine in reducing liver cancer risk. However, the authors recognize the limitations in the study due to residual confounding and concerns about the accuracy and completeness of the data, so further investigations should be done to confirm these effects.

The suggested minor revisions are related to the discussion of the results. Regarding the mechanism of action, it is well stablished that colchicine inhibits tubulin polymerization by binding the colchicine site in tubulin, which is located in the interface between alpha and beta subunits. The authors should better discuss the relationship between the antimitotic effect of colchicine and its effect in liver cancer, which is well documented in literature and could support their findings. 

Author Response

Dearest Reviewer

Your sagacious insights into our manuscript have been received with heartfelt gratitude and have provided a beacon of enlightened guidance as we sought to refine our work. The time and meticulous scrutiny you have graciously bestowed upon our manuscript are sincerely appreciated. Kindly permit us to detail how your esteemed critiques have been woven into our revised submission.

Discussion on Colchicine’s Mechanism:

We have endeavored to enhance our discussion regarding colchicine’s antimitotic effects, particularly its ability to inhibit tubulin polymerization, and how this is intertwined with its potential impact on liver cancer. Your reminder to delve deeper into well-established mechanisms and to intertwine this with relevant literature has been invaluable. We have endeavored to augment our manuscript with additional discussions and citations that spotlight this mechanism, providing a richer context and strengthening the foundation upon which our findings stand.

In every step of our revision, your constructive observations have been our guide, and we have ardently sought to ensure that each aspect of our manuscript now reflects the scholarly rigor and accuracy that befits our scientific community. We humbly submit our revised manuscript for your esteemed perusal and sincerely hope that our amendments align with your expectations and serve to enhance the contribution of our work to the field.

With the deepest respect and gratitude,

Tom XianXiu Chen, MPH-DrPH, Ph.D.

China Medical University Hospital

[email protected]

+886-900-009688

Round 2

Reviewer 1 Report

The changes that are required have been made. This version of the manuscript is accepted.